# Time-Of-Flight ERDA for Depth Profiling of Light Elements

**Keisuke Yasuda** 

Graduate School of Life and Environmental Science, Kyoto Prefectural University, 1 Hangicho, Shimogamo, Sakyo-Ku, Kyoto 606-8522, Japan; yasuda@kpu.ac.jp

**Abstract:** The time-of-flight elastic recoil detection analysis (TOF-ERDA) method is one of the ion beam analysis methods that is capable of analyzing light elements in a sample with excellent depth resolution. In this method, simultaneous measurements of recoil ion energy and time of flight are performed, and ion mass is evaluated. The energy of recoil ions is calculated from TOF, which gives better energy resolution than conventional Silicon semiconductor detectors (SSDs). TOF-ERDA is expected to be particularly applicable for the analysis of light elements in thin films. In this review, the principle of TOF-ERDA measurement and details of the measurement equipment along with the performance of the instrumentation, including depth resolution and measurement sensitivity, are described. Examples of TOF-ERDA analysis are presented with a focus on the results obtained from the measurement system developed by the author.

**Keywords:** TOF-ERDA; ERDA; light elements; depth profiling

---

## 1. Introduction

In recent years, research and development of hydrogen storage alloys, lithium-ion batteries, fuel cells, etc., have become active due to growing interest in energy and environmental issues. Light elements such as hydrogen and lithium play a vital role in these materials. In addition, miniaturization of materials has increased the importance of research on thin films. Thin films with a thickness of several nm play a key role in semiconductor devices and hard disk head materials [1,2].

Ion beam analysis is a method for elemental analysis and depth profiling using ion beams with an energy range of several MeV [3]. The time-of-flight elastic recoil detection analysis (TOF-ERDA) method is one of the ion beam analysis methods specializing in light element analysis and offers acceptable mass separation and good depth resolution [4–9]. In this method, simultaneous measurements of recoil ion energy and time of flight are performed, and ion mass is evaluated. This in turn enables particle discrimination measurements. The energy of recoil ions is calculated from TOF, which gives better energy resolution than conventional silicon semiconductor detectors (SSDs). Moreover, a range foil, which is mandatory in conventional stopper-foil ERDA measurements, is not required. This is advantageous for excellent depth resolution measurements. This paper outlines the principles and measurement techniques of the TOF-ERDA method, and introduces the measurement system developed by the author and its performance along with those of other TOF-ERDA systems. An example of the application is also presented.

## 2. Principle of ERDA

In ERDA, recoil ions are detected when the swift incident ions are elastically scattered by atoms in the sample. It is appropriate for element analysis and depth profiling of light elements, including hydrogen [3,10,11]. It is also possible to analyze heavy elements in addition to light elements by using heavy ion beams such as iodine and gold.

In the elastic scattering between incident ions and target atoms, recoil atoms are emitted at $\phi \leq 90°$, where $\phi$ is the recoil angle. Hence, the recoil ions are detected at a forward angle with respect to the beam axis. In a typical ERDA, an ion beam is injected into the sample at a shallow angle and a recoil ion emitted from the same surface is detected, as shown in Figure 1. When the collision occurs at the sample surface, the kinetic energy of the recoil atom, $E_r$, is assessed using Equation (1).

$$E_r = k_r E_0,$$
$$k_r = \frac{4 M_p M_r}{(M_p + M_r)^2} \cos^2 \phi \tag{1}$$

where $E_0$ is the incident energy and $M_p$ and $M_r$ are masses of the incident ion and recoil atom, respectively. $k_r$ is called *k-factor*. If the collision occurs inside the sample, the incident ions and recoil atom lose their kinetic energy as they move through the sample. Therefore, the kinetic energy of the recoil atom at the sample surface becomes lower than $E_r$. When the incident ion collides with the target atom at the depth $x$ perpendicular to the surface, as shown in Figure 1, the kinetic energy of the recoil atom at the sample surface, $E_2(x)$, is expressed as follows:

$$E_2(x) = k_r\left(E_0 - \Delta E_p\right) - \Delta E_r \tag{2}$$

where $\Delta E_p$ and $\Delta E_r$ are the energy losses of incident and recoil ions in the sample, respectively. These quantities are expressed using the stopping power for the incident and recoil ions as follows:

$$\Delta E_p = \int_0^{\frac{x}{\sin \alpha}} \frac{dE}{dX}\left(E_p(X)\right)dX, \quad \Delta E_r = \int_0^{\frac{x}{\sin \beta}} \frac{dE}{dX}(E_r(X))dX. \tag{3}$$

where $E_p(X)$ and $E_r(X)$ stand for energies of incident and recoil ions in the sample, respectively. Defining $\langle S_p \rangle$ and $\langle S_r \rangle$ by Equation (4), the energy loss at depth x is represented by Equation (5):

$$\langle S_p \rangle = \left(\int_0^{\frac{x}{\sin \alpha}} \frac{dE}{dX}\left(E_p(X)\right)dX\right) / \left(\int_0^{\frac{x}{\sin \alpha}} dX\right),$$
$$\langle S_r \rangle = \left(\int_0^{\frac{x}{\sin \beta}} \frac{dE}{dX}(E_r(X))dX\right) / \left(\int_0^{\frac{x}{\sin \beta}} dX\right). \tag{4}$$

$$\Delta E_p = x\frac{\langle S_p \rangle}{\sin \alpha}, \quad \Delta E_r = x\frac{\langle S_r \rangle}{\sin \beta}. \tag{5}$$

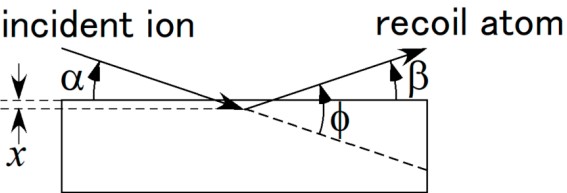

**Figure 1.** Principle of ERDA. $\phi$, $\alpha$ and $\beta$ are recoil angle, incident angle, and exit angle, respectively. $x$ is the depth where the incident ion collides with the target atom.

By substituting Equation (5) in Equation (2):

$$E_2(x) = k_r E_0 - x\left(k_r\frac{\langle S_p \rangle}{\sin \alpha} + \frac{\langle S_r \rangle}{\sin \beta}\right), \tag{6}$$

where $\alpha$ and $\beta$ are incident and exit angles, respectively, as shown in Figure 1. Equation (6) shows the relationship between measured recoil energy and depth. This relationship allows us to determine the depth, where the collision occurs, from the measured recoil energy. For the evaluation of the stopping power, SRIM code is widely used [12].

The concentration of an element at the depth of $x$ is expressed as below:

$$\rho(x) = \left(-\frac{dE_2}{dx}\right)\frac{Y(E_2)}{\sigma_r N_b \Delta\Omega\varepsilon},$$

(7)

where $Y(E_2)$ is number of recoil ions detected with energy $E_2$ per unit energy, $\sigma_r$ is the differential recoil cross section, $N_b$ is the number of incident ions, $\Delta\Omega$ is the solid angle and $\varepsilon$ is the detection efficiency. $(-dE_2/dx)$ in Equation (7) is derived from Equation (6). When the stopping powers, $S_p$ and $S_r$, do not change significantly within the measured depth range, $(-dE_2/dx)$ is expressed as below:

$$-\frac{dE_2}{dx} = k_r\frac{\langle S_p\rangle}{\sin\alpha} + \frac{\langle S_r\rangle}{\sin\beta}.$$

(8)

The evaluation of $\sigma_r$ depends on the closest distance between the two colliding nuclei, $d$, which depends on the incident energy and atomic numbers of incident and target atoms. If $d$ lies roughly between the sum of the nuclear radius and K-shell radius, $\sigma_r$ can be expressed using Rutherford's formula.

$$\sigma_r = \left(\frac{Z_p Z_r e^2}{2E_p}\right)^2\left(\frac{M_p + M_r}{M_r}\right)^2\frac{1}{\cos^3\phi},$$

(9)

where $Z_p$ and $Z_r$ are atomic numbers of the incident ion and target atom, respectively, and $e$ is the elementary charge. If $d$ is smaller than the sum of nuclear radii, $\sigma_r$ is affected by nuclear force and deviates from the Rutherford cross section. This is essential for the hydrogen analysis due to the small Coulomb barrier. The non-Rutherford cross section can be obtained from experimental data [13] or calculation [14,15]. If $d$ is greater than the K-shell radius of incident or target atoms, the screening effect must be considered. The screening effect is important for low-energy TOF-ERDA measurements using a heavy ion beam of several MeV. Calculating the recoil cross section incorporating the screening effect can be difficult and its accuracy can be a problem for quantitative analysis [6].

In the typical ERDA measurement, ions of several MeV of energy accelerated by an electrostatic accelerator are used as the incident ions, and recoil ions are detected with charged particle detector(s). Scattered ions as well as recoil ions can go in the direction of the detector. In this case, discrimination between scattered and recoil ions is necessary in the ERDA measurement. The most widely used method for the discrimination is to set a foil before the detector. The foil thickness, typically several μm, is determined so that only recoil atoms can penetrate. This method is called stopper-foil ERDA. In the stopper-foil ERDA, ions with an atomic number greater than that of a measured element are used as incident ions. By choosing a foil with appropriate thickness, scattering ions can be stopped in the foil and only recoil ions can be incident on the detector as the range of the recoil ions in the material is longer than that of the scattered ions. A polymer or a metal film with a thickness of several μm is used as the stopper foil. Simultaneous measurement of multiple elements is difficult in the stopper-foil ERDA because the ion species of the recoil particles cannot be identified. In addition, depth resolution deteriorates significantly due to the energy straggling when the recoil ions pass through the stopper foil. When analyzing hydrogen using helium ions as the incident beam, the achievable depth resolution is about 10 nm [16].

## 3. Principle and Measurement System of the TOF-ERDA

### 3.1. Overview

In the TOF-ERDA, energy and time of flight (TOF) of the recoil ion are measured simultaneously. Mass of the recoil ion, $m$, can be determined as follows:

$$m = 2E\left(\frac{TOF}{l}\right)^2$$

(10)

where $E$ is the energy of the ion and $l$ is the flight path. Since ion species are identified by the mass, multiple elements can be analyzed in a single measurement. Figure 2 shows a schematic of a typical TOF-ERDA set-up. Two time detectors and one energy detector are employed. TOF is measured by two time detectors and energy is measured by an energy detector. There are several types of time detectors; however, the Busch type is widely used [17–19]. A schematic of the Busch type time detector is shown in Figure 3. It consists of a micro channel plate (MCP) detector, carbon foil and electrostatic mirror grids. Secondary electrons generated as ions pass through the carbon foil are reflected by the mirror grid and then detected by the MCP detector [19]. The thickness of the carbon foil is about 1 to 10 $\mu g/cm^2$, which corresponds to about 1/100 to 1/1000 of the thickness of the stopper foil. The influence of the energy straggling is small due to the small thickness of the carbon foil, so that measurements with better depth resolution can be performed. In many facilities, a silicon semiconductor detector (SSD) has been used as the energy detector, however, a gas ionization chamber (GIC) which provides superior energy resolution for low-energy heavy ions, has been used in some cases [4,20]. The recoil energy is calculated using the TOF value. In TOF-ERDA measurements, heavy ions such as chlorine, copper, bromine, iodine and gold are often used as the incident beam [4–9]. However, light ions such as helium can also be used [21,22] in measurements. In this case, cross section data are required for quantitative analysis [23]. On the other hand, in the case of the heavy ion beam, the scattering cross section can be calculated by Rutherford's formula because of the strong Coulomb interaction and the almost total absence of nuclear force effects. However, when the incident energy is low, the screening effect should be taken into account.

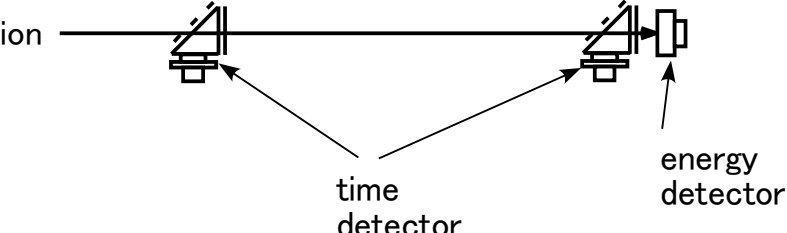

**Figure 2.** Schematic of TOF-ERDA set-up.

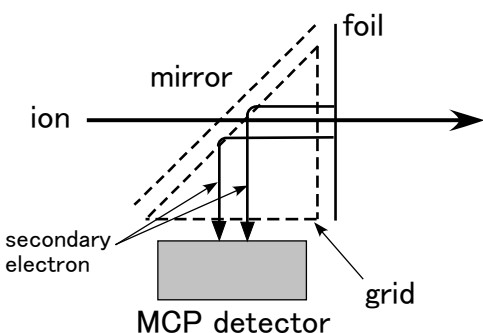

**Figure 3.** Schematic of the Busch type time detector.

### 3.2. Measurement Set-Up

After briefly introducing some of the TOF-ERDA measurement systems in operation, a TOF-ERDA measurement system installed at the Wakasa Wan Energy Research Center (WERC) is described in detail as an example [24].

The reported specifications of typical TOF-ERDA systems (Zurich [4], Zagreb [5], Leuven/IMEC [6] and Jyväskylä [7]) are summarized in Table 1. Detection angles are around 40° in Zagreb, Leuven/IMEC and Jyväskylä, and they can be moved in the angular range between 26° and 46° in Zurich. The flight length ranges between 520 mm and 950 mm. The Busch type detectors are used as a time detector in all cases. For the foil of the first time detector, a diamond-like carbon (DLC) film of 0.5 $\mu g/cm^2$

thickness [4,5] or carbon foil of 3 µg/cm$^2$ thickness [6,7] are used. In Zagreb and Jyväskylä, a thin layer of LiF [5] or Al$_2$O$_3$ [6] is coated on the foil for the improvement of the detection efficiency for light elements. Solid angles range between 0.11 msr and 0.44 msr. An SSD is utilized for the energy detector in Zagreb, Leuven/IMEC and Jyväskylä, whereas a GIC is used in Zurich for the improvement of mass separation. The mass resolution *M*/Δ*M* of ~40 is reported for the detection of $^{39}$K in the case of a 12 MeV $^{127}$I beam incident [4].

**Table 1.** Summary of typical time-of-flight elastic recoil detection analysis (TOF-ERDA) equipment specifications.

| Facility | Recoil Angle | Flight Length (mm) | T1 Foil | T2 Foil | Solid Angle (msr) | Energy Detector |
|---|---|---|---|---|---|---|
| Zurich [4] | 26°–46° | 950 | 0.5 µg/cm$^2$ Diamond-like carbon (DLC) | 3.0 µg/cm$^2$ Carbon | 0.26 | Gas ionization chamber (GIC) |
| Zagreb [5] | 37.5° | 520 | 0.5 µg/cm$^2$ DLC + LiF | | 0.11 | Silicon semiconductor detector (SSD) |
| Leuven/IMEC [6] | 38.5° | 570 | 3.0 µg/cm$^2$ Carbon | 10.0 µg/cm$^2$ Carbon | 0.44 | SSD |
| Jyväskylä [7] | 41.3° | 623 | 3 µg/cm$^2$ Carbon + 1 nm Al$^2$O$^3$ | 7 µg/cm$^2$ Carbon | 0.29 | SSD |
| WERC [22] | 40° | 589 | 3 µg/cm$^2$ Carbon | 10 µg/cm$^2$ Carbon | 0.09 | SSD |

The specification of the TOF-ERDA system at the WERC is also presented in Table 1. This system is employed for light element analysis of various materials such as Li-ion batteries and ceramics [25–27]. A schematic view of the TOF-ERDA set-up is illustrated in Figure 4. An ion beam accelerated by a 5 MV tandem accelerator is used as the incident beam [28]. The typical beam size is about $1 \times 1$ mm$^2$, which is defined by two sets of slit systems installed upstream of the scattering chamber. A scattering chamber used for common ion beam analysis (IBA) measurements such as Rutherford backscattering spectroscopy (RBS) and stopper-foil ERDA is used. A TOF pipe is connected to the 40° port of the scattering chamber and consequently the detection angle is 40° with respect to the beam direction. Samples are attached to a 5-axis goniometer. The measurement system is exhausted by a turbo molecular pump and the pressure during the measurement is kept at about $5 \times 10^{-5}$ Pa.

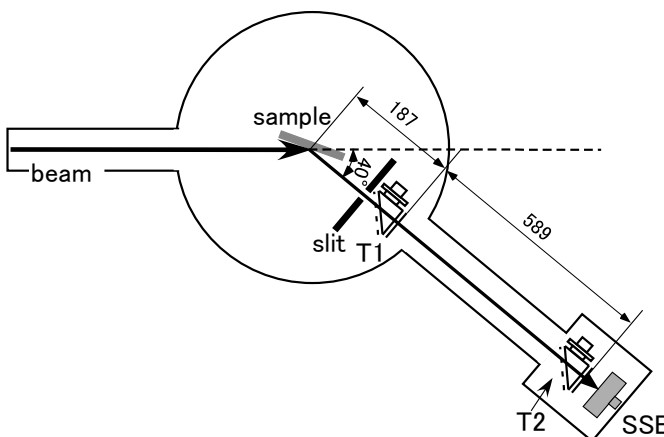

**Figure 4.** Schematic view of TOF-ERDA measurement set-up at WERC.

The detection system consists of two time detectors and one energy detector, as mentioned in the previous subsection. The first time detector (T1) is placed on a turn table in the scattering chamber, whereas the second time detector (T2) is installed in the small chamber connected to the TOF pipe. Two-stage MCP detectors (HAMAMATSU F4655-12) are used as the secondary electron detectors in the time detectors. Typical operation voltages of the MCP detectors are 0 V for MCP in, +2200 V for MCP out and +2300 V for the anode. The electric potentials of the foil and mirror are −1000 V and −2000 V, respectively. The diameters of the carbon foils, which make sensitive areas of the time detectors, are 2 mm for T1 and 10 mm for T2. The thicknesses of the carbon foils are 3 µg/cm$^2$ for T1 and 10 µg/cm$^2$ for T2. The distance between T1 and T2 is 589 mm. The solid angle of the detection system is 0.090 msr, which is defined by the sensitive area of T1. Because of the wires in the acceleration

and mirror grids, the transmission is 77%, and 5% of the ions entering the first detector cannot reach the second detector due to the beam spread in the case of an incident angle of 20 degrees. As a result, the effective solid angle is ~0.066 msr. A silicon surface barrier (SSB) detector (ORTEC BF-023-300-60) is used as the energy detector. It is located just behind T2.

An overview of the signal processing circuit is drawn in Figure 5. Signals from time detectors were led to constant fraction discriminators (CFDs, ORTEC 935). Threshold voltages of the CFDs were set to their minimum ($\approx -18$ mV). Logic signals from the CFD outputs are fed into a time-to-amplitude converter (TAC, ORTEC 567) for the TOF measurements. The count rate of T1 is several tens of times higher than that of T2. This higher count rate is mainly caused by the secondary electrons from the sample entering directly into the T1 MCP detector. In order to reduce the deadtime of the TAC due to the high count rate of the T1 MCP detector, the signal from the T2 MCP detector is used to start the TAC. The delayed signal from the T1 MCP detector then stops the TAC. The delay times in the MCP signals were tuned so that the signal from the T1 arrived during the time window of the TAC (500 ns). The trigger signal of the data acquisition system is generated by the signal from the SSB detector. The pulse heights of the signals from the SSB detector and TAC are recorded by a list-mode data acquisition system.

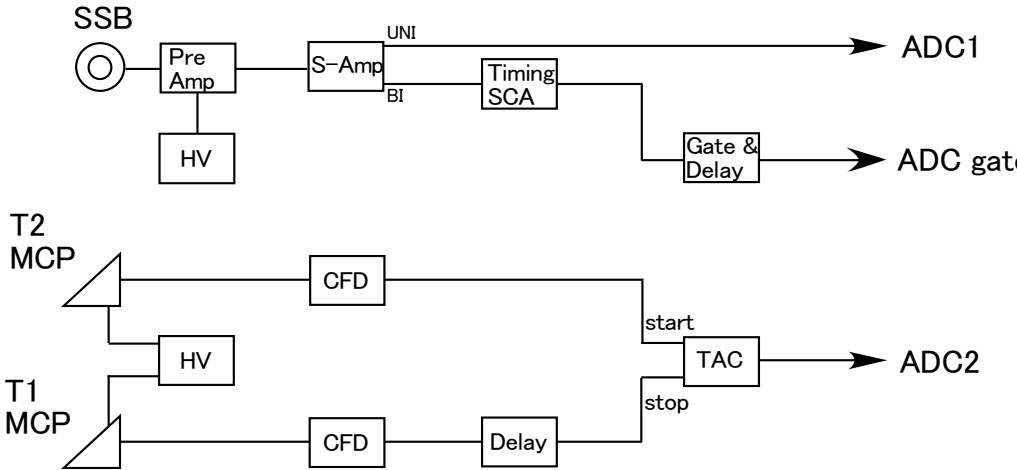

**Figure 5.** Overview of the signal processing circuit of the WERC system.

## 4. System Performance

In this chapter, the basic performances of our and typical TOF-ERDA measurement systems are described.

### 4.1. Time Resolution

The time resolution of the TOF measurement is significant as it greatly affects the mass resolution and depth resolution of the TOF-ERDA measurement. Time resolutions of 154 ps at Jyväskylä [8], 170 ps at Zagreb [5], 218 ps at Montreal [9], 400 ps at Zulich [4] and 540 ps at Johannesburg [7] have been reported. The Leuven/IMEC group has shown ion species dependence of the time resolution, 550 ns for heavy ions, 900 ps for He and 1 µs for H. Measurements of time resolution have been performed by measuring recoil ions [4,7], scattered ions [5,8] or $\alpha$-rays from a radioactive source [9].

In our system, the time resolution was measured by detecting $^4$He ions scattered by an Au thin film. As the incident beam, 5.7 MeV $^4$He$^{2+}$ ions were used. The Au layer with 2.1 µg/cm$^2$ thickness, which was deposited on a silicon wafer, was used as the target. A TOF spectrum of He ions scattered

by the Au layer is shown in Figure 6. The peak width, Δt, was determined to be 390 ps (full width at half-maximum, FWHM) by fitting with a Gaussian distribution function. Δt is expressed as follows:

$$\Delta t^2 = \Delta t_{int}^2 + \frac{ml^2}{8E^3}\Delta E^2 + \frac{m}{2E}\Delta l^2 \tag{11}$$

where $\Delta t_{int}$ is the intrinsic time resolution of the TOF measurement system, $\Delta l$ is the variation in flight path length, $\Delta E$ is the energy spread and $m$, $l$, $E$ are mass, flight length and energy of the detected particle, respectively. The contributions from the second and third terms of Equation (11) are less than 40 ps, which are negligibly small compared to the measured value of 390 ps. Therefore, the width of the peak can be considered as the intrinsic time resolution of the measurement.

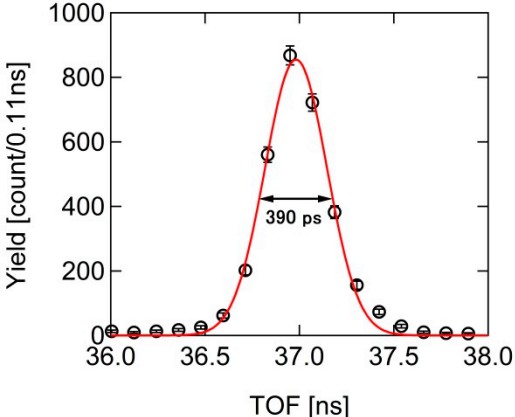

**Figure 6.** TOF spectrum of scattered He ions by a Au thin layer with 2.1 µg/cm$^2$ thickness obtained with a 5.7 MeV $^4$He beam. A Gaussian fit is also plotted. Time resolution was estimated to be 390 ps [24].

*4.2. Detection Efficiency*

It is imperative to accurately determine the detection efficiency of the measurement system for quantitative analysis of the TOF-ERDA measurement. The detection efficiency of the SSB detector for charged particles can be considered as 100%. On the other hand, the efficiency of a time detector is less than 100%, and depends on the ion species and energy [29]. The detection efficiency of the time detector is greatly affected by the number of secondary electrons generated in the foil. It is positively correlated with the stopping power of the recoil particles in the carbon foil. Therefore, the detection efficiency of light ions, such as hydrogen and helium, is less than 1. On the other hand, detection efficiencies of nearly 90% to 100% have been reported for Li and heavier ions [5,6,8]. In some facilities, carbon or DLC foils coated with LiF [5] or Al$_2$O$_3$ [8] have been used to improve detection efficiency for light ions such as hydrogen and helium, and detection efficiencies of 10% to 85% for hydrogen and detection efficiencies of more than 90% for helium below 2 MeV have been reported [5,8].

The detection efficiency of the time detector in our system was evaluated experimentally [25]. Measurements were carried out using the TOF-ERDA measurement set-up. $^1$H, $^4$He and $^{12}$C ions from the tandem accelerator were irradiated to a thick gold target with 200 µm thickness and scattered ions were detected with the measurement system. In order to reduce the number of accidental coincidence events, beams with an intensity of less than 0.1 particle nA were used. Count rates of the SSD and T1 were 35–45 cps and 1800–3300 cps for H and He measurements, and 30–60 cps and 6500–7000 cps for C measurements, respectively. The detection efficiency, $\varepsilon$, was determined using the following formula:

$$\varepsilon = \frac{Number\ of\ coincident\ events\ of\ TOF\ and\ SSB}{Number\ of\ SSB\ events} \tag{12}$$

Since TOF is measured only when the signal is generated by both time detectors, the multiplied efficiency of the two time detectors is obtained. The results are shown in Figure 7. It is realized that if the energies are the same, the detection efficiency increases with the atomic number of the ion. The reason lies in the fact that the number of secondary electrons emitted from the foil has a positive correlation with the stopping power of the foil against ions. At the same energy, the stopping power increases with the atomic number of the particle, and hence the number of the secondary electrons emitted increases. This in turn increases the detection efficiency [29]. The detection efficiency for the carbon ions with energy between 1 MeV and 10 MeV was approximately 85%. On the other hand, the efficiency for helium ions with energy below 6 MeV was between 60% and 20%, and the efficiency for hydrogen ions with energy below 2 MeV was less than 30%.

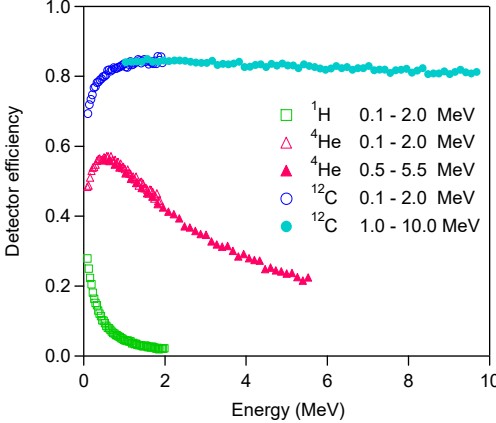

**Figure 7.** Detection efficiency for $^1$H, $^4$He and $^{12}$C ions [25].

Figure 8 shows the detection efficiencies of T1 obtained by measurements using an 8 MeV $^{12}$C beam and a 12 MeV $^{63}$Cu beam as a function of the electric stopping power of the ions in carbon obtained with SRIM-2008 [12]. The detection efficiency approaches 1 as the electric stopping power increases. On the other hand, other facilities have reported detection efficiencies of more than 97% with two detectors for carbon at energies of 1–5 MeV. The reason for the lower detection efficiency of our measurement system may be due to the low output of MCPs compared to the threshold of CFDs. In order to avoid the increase in dark noise and discharge, the applied voltage of MCP out is operated at 2200 V, which is 300 V lower than the maximum voltage, which is considered to reduce the detection efficiency of the time detector.

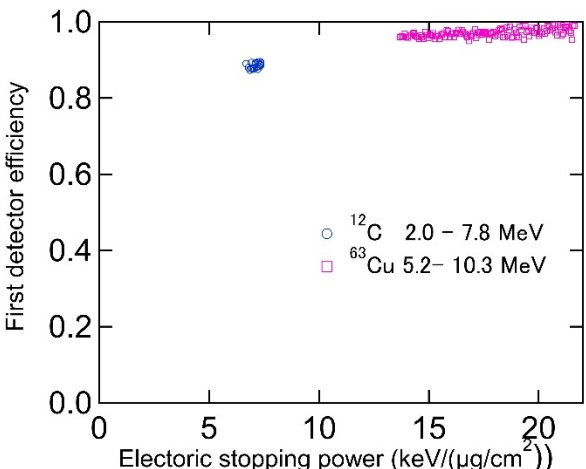

**Figure 8.** Detection efficiency of T1 for $^{12}$C and $^{63}$Cu ions as a function of the electric stopping power.

### 4.3. Depth Resolution

Depth profiling is one of the features of the TOF-ERDA method, and is particularly useful for analyzing thin films. Depth resolution is essential in the depth profiling. In TOF-ERDA measurements, surface depth resolutions of 1.5 nm for O in $TiO_2$ and 5–5.5 nm for O in $SiO_2$ are reported [4–6]. In this subsection, experimental results of the depth resolution measurement obtained with our setup are presented.

Depth resolution at the carbon surface was measured using $^4$He and $^{63}$Cu beams. Silicon wafers coated with a carbon thin layer of about a 50 nm thickness, which was deposited by an arc discharge method, were used as samples. The TOF spectrum of the recoil carbon obtained with a 5 MeV $^{63}$Cu beam and incident/exit angles of 35°/5° is shown in Figure 9. The surface edge of the TOF spectrum was fitted using an error function, and the time broadening, $\delta t$, was obtained. The energy of the carbon ion is calculated from the TOF by the following formula:

$$E = \frac{m}{2}\left(\frac{l}{TOF}\right)^2 \tag{13}$$

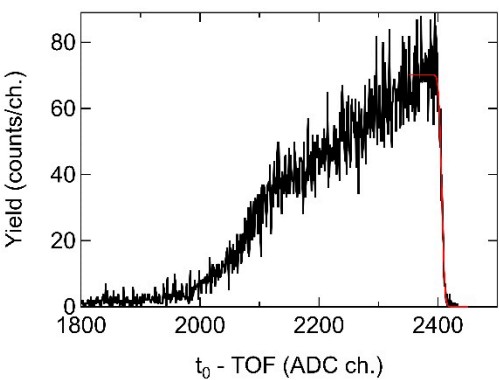

**Figure 9.** TOF spectrum of recoil carbon obtained with a 5.0 MeV $^{63}$Cu beam and incident/exit angles of 35°/5°. Broadening at the surface edge, $\delta t$, was evaluated by fitting with the error function. The equation for the relationship between time and Analog–to–digital converter (ADC) channel was obtained by the calibration measurement using a time calibrator (ORTEC 462).

Therefore, the following relationship between the energy broadening, $\delta E$, and $\delta t$ holds [3]:

$$\frac{\delta E}{E} = 2\frac{\delta t}{TOF} \tag{14}$$

From Equations (13) and (14), $\delta E$ can be calculated as follows:

$$\delta E = \sqrt{\frac{8E^3}{ml^2}}\delta t \tag{15}$$

The depth resolution at the surface, $\delta x$, was deduced from the energy broadening, $\delta E$, using the following expressions:

$$\delta x = \frac{\delta E}{[S]_{p,r}} \tag{16}$$

$$[S]_{p,r} = k_r\frac{S_p(E_0)}{\sin\alpha} + \frac{S_r(E_2)}{\sin\beta} \tag{17}$$

where $\delta E$ is the energy broadening, $S_p$ and $S_r$ are the stopping powers of the incident and recoil ions, $E_0$ and $E_2$ are the energies of incident and recoil ions, $\alpha$ and $\beta$ are incident and exit angles and $k_r$ is the

kinematical factor, defined as $E_2/E_0$. The stopping powers were evaluated using the SRIM2008 code [12]. The dependence of the depth resolution on the incident/exit angles is explained by Equations (16) and (17). In Equation (16), the energy spread $\delta E$ is independent of the incident/exit angles, whereas $[S]_{p,r}$ depends on the incident/exit angles, $\alpha$ and $\beta$, as shown in Equation (17), and increases as either of them becomes smaller. Therefore, the depth resolution improves as the incident or exit angle decreases. The measurement was carried out with a small exit angle in order to avoid increasing the irradiated area on the sample. The experimental results of the depth resolution are plotted as a function of the incident energy and the incident/exit angles in Figure 10. It shows that the depth resolution became better for lower incident energies and a smaller exit angle. A depth resolution of 1.3 ± 0.1 nm was obtained with a 2 MeV $^4$He beam and incident/exit angles of 35°/5°, while 1.6 ± 0.2 nm was obtained with a 5 MeV $^{63}$Cu beam and incident/exit angles of 35°/5°, where 2.253 g/cm$^3$ is assumed to be the density of the carbon layer. Calculations are also shown with different colors. The calculation includes the effects of kinematical broadening, time resolution of the TOF measurement, straggling in the carbon foil and non-uniformity of the carbon foil thickness. The straggling in the carbon foil was estimated with Bohr's formula [3], and the non-uniformity of the carbon foil thickness was assumed to be 30%. The calculations indicate that the kinematical broadening mainly contributes to the depth resolution, and TOF time resolution also makes a significant contribution especially at higher incident energy. The depth resolution at the surface of the sample was discussed here, however, the depth resolution inside the sample is important in actual measurements. Giangrandi et al. reported a depth resolution of ~10 nm at a depth of 20 nm and ~16 nm at a depth of 50 nm in the measurement of O in $SiO_2$ using a $^{35}$Cl beam [6].

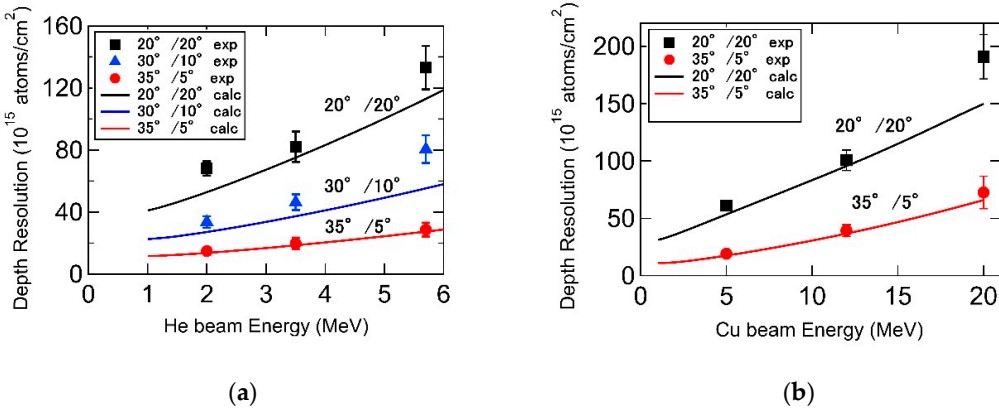

**Figure 10.** Depth resolution for the carbon surface as a function of incident energy for (**a**) $^4$He and (**b**) $^{63}$Cu. Solid lines show calculation results. Black, blue and red correspond to the incident/exit angles of 35°/5°, 30°/10° and 20°/20°, respectively.

*4.4. Sensitivity*

Measuring the instrument sensitivity is essential for determining analytical capabilities. Sensitivity in the TOF-ERDA measurement depends on conditions such as measurement element, matrix, incident ion species, incident energy and so on. In the present paper, the sensitivity for oxygen in silicon using $^{12}$C and $^{63}$Cu beams is discussed.

The sample for the sensitivity measurements was prepared by implanting oxygen ions with 50 keV energy on silicon wafers. The oxygen areal density in the sample, $\sigma_{sample}$, was $1.2 \times 10^{17}$ cm$^{-2}$, which was confirmed by a RBS measurement with 1.5 MeV He$^+$ ions. In the TOF-ERDA measurements, 12 MeV Cu$^{11+}$ ions and 8 MeV C$^{3+}$ ions were used as the incident beams. The beam currents were 0.09~0.15 particle nA for the Cu$^{11+}$ beam and about 3.3 particle nA for the C$^{3+}$ beam, which are considered as typical values for the TOF-ERDA measurement with the introduced set-up. The total measurement time was 2 h for each case. Pure silicon wafers were measured for the same amount of time for background evaluation. Count rates of the detectors as well as the beam currents for each measurement are shown in Table 2.

**Table 2.** Count rates of detectors and beam current for each measurement.

| Beam | Sample | Count Rate (cps) | | | Beam Current (Particle nA) |
|---|---|---|---|---|---|
| | | T1 | T2 | SSD | |
| 12 MeV $^{63}$Cu | implant | $1.5 \times 10^4 \sim 2.5 \times 10^4$ | $1.7 \times 10^2 \sim 2.6 \times 10^2$ | 60~90 | 0.09~0.15 |
| | pure | $1.0 \times 10^4 \sim 1.8 \times 10^4$ | $1.4 \times 10^2 \sim 2.3 \times 10^2$ | 50~70 | 0.09~0.14 |
| 8 MeV $^{12}$C | implant | $3.0 \times 10^4$ | $7.7 \times 10^2$ | $2.9 \times 10^2$ | 3.3 |
| | pure | $3.2 \times 10^4$ | $8.6 \times 10^2$ | $3.7 \times 10^2$ | 3.3 |

Figures 11 and 12 show the TOF versus energy plots of (a) oxygen-implanted silicon wafer and (b) pure silicon wafer. Events in the oxygen gate, indicated by dashed lines in Figures 11 and 12, were used for the evaluation of the detection limit. Recoil oxygen events from the surface oxide layer were excluded in the analysis.

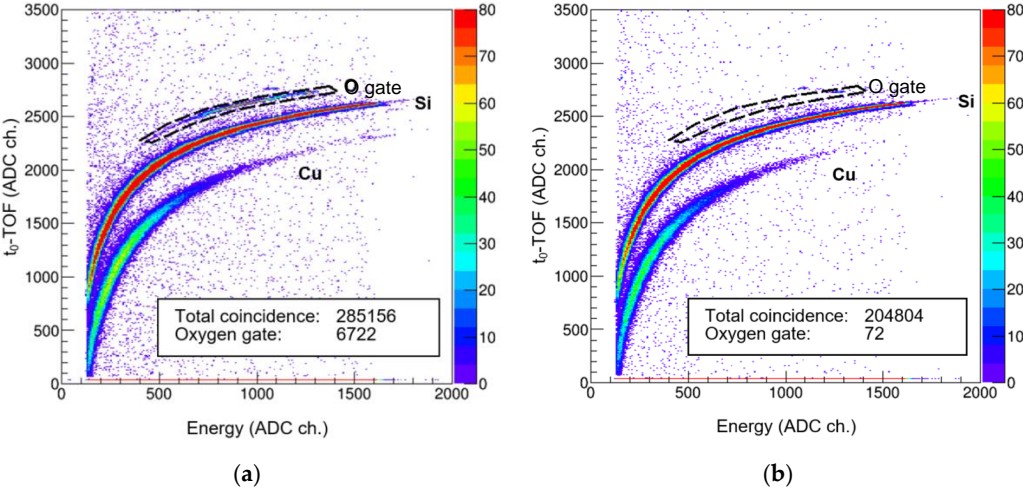

| (a) | (b) |

**Figure 11.** TOF versus energy plots obtained using 12 MeV $^{63}$Cu beam for (**a**) oxygen-implanted silicon wafer and (**b**) pure silicon wafer. The measurement time for both was 2 h. Areas shown by dashed lines indicate gates for the selection of oxygen events. The total number of TOF-E coincident events and oxygen events are also shown in the figures.

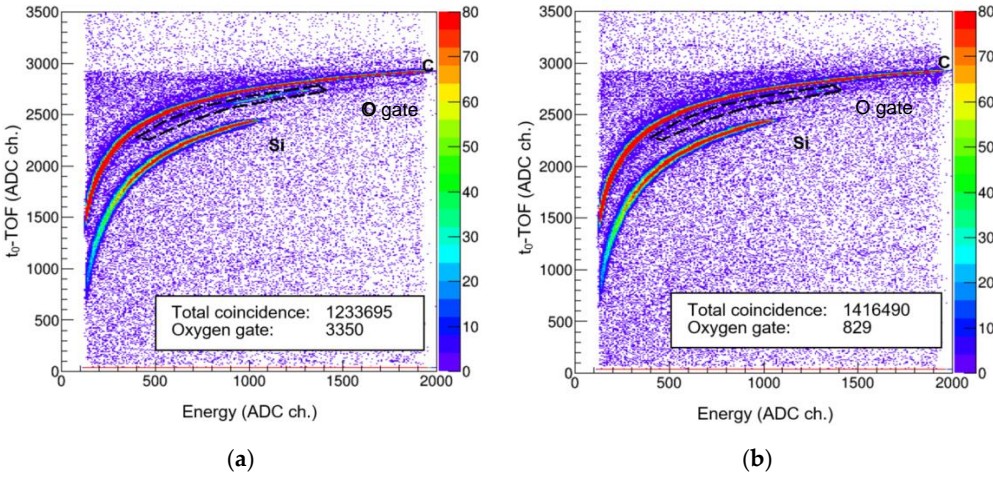

| (a) | (b) |

**Figure 12.** TOF versus energy plots obtained using 8 MeV $^{12}$C beam for (**a**) oxygen-implanted silicon wafer and (**b**) pure silicon wafer. The measurement time for both was 2 h. Areas shown by dashed lines indicate gates for the selection of oxygen events. The total number of TOF-E coincident events and oxygen events are also shown in the figures.

The detection limit is evaluated as follows. Let the number of recoil oxygen and background events be $Y$ and $Y_{BG}$, respectively. When these values satisfy Equation (18), oxygen is considered to have been detected.

$$Y \geq 3 \sqrt{Y_{BG}}, \tag{18}$$

The detection limit is reached when an equal sign is established. When $\sigma_{sample}$ and $Y_{sample}$ are the areal density and the number of events for the sample, $\sigma_l$ and $Y_l$, are the areal density and the number of events at the detection limit, the following relation holds:

$$Y_l = \frac{\sigma_l}{\sigma_{sample}} Y_{sample} \tag{19}$$

Since $Y_l = 3 \sqrt{Y_{BG}}$ is obtained from Equation (18), substituting this into Equation (19) yields the following:

$$\sigma_l = \frac{\sigma_{sample}}{Y_{sample}} 3 \sqrt{Y_{BG}}, \tag{20}$$

The detection limits obtained using Equation (20) are shown in Figure 13 as a function of the number of incident ions. The detection limit becomes lower as the number of incident ions increases, indicating that the sensitivity increases as the number of incident ions increases. The detection limit in the 2 h measurements were $(3.8 \pm 0.2) \times 10^{14}$ cm$^{-2}$ for the Cu beam and $(3.7 \pm 0.1) \times 10^{15}$ cm$^{-2}$ for the C beam, indicating that the sensitivity for the Cu beam was 9.7 times higher.

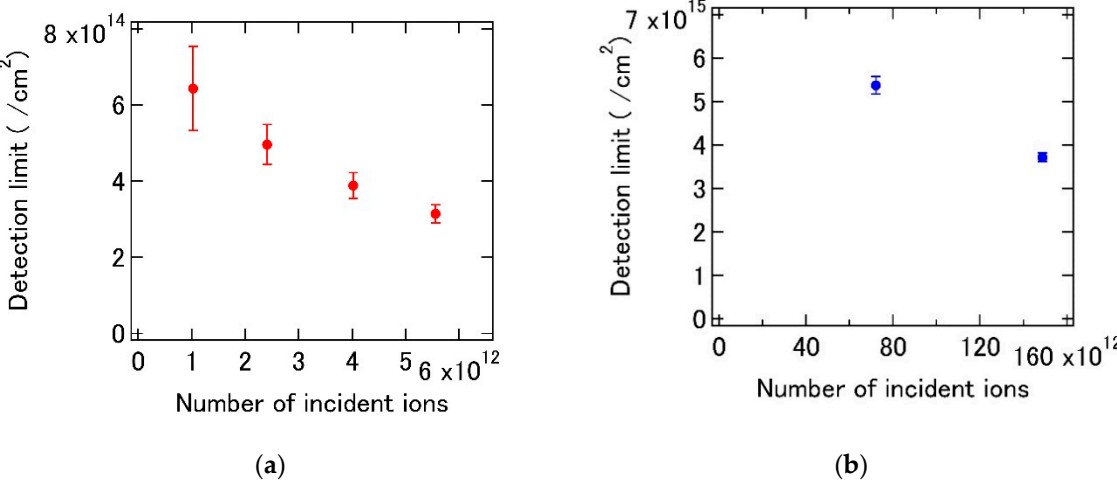

(**a**)  (**b**)

**Figure 13.** Sensitivity for 12 MeV $^{63}$Cu (**a**) and 8 MeV $^{12}$C (**b**). Horizontal and vertical axes show the number of incident ions and detection limit, respectively. The points with the largest number of incident ions correspond to 2 h of measurement.

The ratio of the number of recoil oxygen events with the same number of incident ions, $Y(Cu)/Y(C)$, was $76.6 \pm 1.7$, which was consistent with the ratio of the Rutherford recoil cross section, 82.7, within 7%. The ratio of the number of background events for the same number of incident ions, $BG(Cu)/BG(C)$, was $2.2 \pm 0.3$. From Equation (20), the ratio of the detection limit for the same number of incident ions, $\sigma_l(C)/\sigma_l(Cu)$, is expressed as follows:

$$\frac{\sigma_l(C)}{\sigma_l(Cu)} = \frac{Y(Cu)}{Y(C)} \sqrt{\frac{BG(C)}{BG(Cu)}}. \tag{21}$$

Substituting the above values into this, we obtained $\sigma_l(C)/\sigma_l(Cu) = 51 \pm 3$. This means that the sensitivity of the Cu beam was 51 times higher than that of the C beam for the same number of incident ions.

The primary origins of the background are the accidental coincidence events of T1 and T2. In the case of the C beam, the background contribution from scattered carbon events is also significant. An accidental coincidence event rate, $R_{acc}$, is approximately proportional to the product of the count rates of T1 and T2 [30]. A rough estimation of the contribution to the background from accidental coincidence and contamination of scattered carbon events in the case of carbon beams was performed. The ratio of $R_{acc}$ between the Cu beam and C beam, $R_{acc}(Cu)/R_{acc}(C)$, is estimated to be $\approx 0.1$ from the values in Table 2. From this, the ratio of the number of events in the accidental coincidence is $\approx 3$ in the case of the same number of incident particles. The accidental coincidence events within the oxygen gate are the background. Since the gates are the same for both beam measurements,

$$\frac{BG_{acc}(Cu)}{BG_{acc}(C)} \approx 3 \tag{22}$$

is obtained, where $BG_{acc}$ is the number of background events due to the accidental coincidence. If $BG_{scat}(C)$ is defined as the number of background events due to the inclusion of scattered carbon events in the carbon beam measurement, the ratio of background events, BG(Cu)/BG(C), is expressed as follows:

$$\frac{BG(Cu)}{BG(C)} = \frac{BG_{acc}(Cu)}{BG_{acc}(C) + BG_{scat}(C)} \approx 2.2 \pm 0.3. \tag{23}$$

From Equations (22) and (23), $BG_{scat}(C)/BG_{acc}(C) \approx 0.3$ is evaluated. This indicates that the background from scattered carbon events is about 30% of the background from the accidental coincidence. As mentioned above, the count rate of the accidental coincidence is proportional to the product of the count rates of T1 and T2, and these depend on the beam intensity. This means that $BG_{acc}$ depends on the beam intensity. The sensitivity can be improved by reducing the beam intensity and taking a long measurement.

The sensitivity for a 2 h measurement using the Cu beam is equivalent to 760 ppm. On the other hand, a sensitivity of 1–10 ppm has been reported for ΔE-E ERDA measurements, for example [31,32], and a similar sensitivity is expected to be obtained by TOF-ERDA systems in operation. For the TOF-ERDA measurement using this system, it is important to improve the detector or optimize the measurement conditions such as beam type and intensity to achieve higher sensitivity.

## 5. Example

As an example of thin film analysis using the TOF-ERDA method, the depth profiling of a $TiO_x$ thin film of about 10 nm thickness, formed on a silicon wafer, is presented. The value of x in $TiO_x$ is determined to be 2.40 by an RBS measurement with 2 MeV He ions. A 5 MeV $^{63}$Cu beam used as the incident beam and the measurement was performed at an incident/exit angle of 35°/5°. A two-dimensional plot of TOF-E and the elemental depth profile evaluated from the plot are shown in Figure 14. In the depth profile, the depth on the horizontal axis is evaluated assuming that the density of the $TiO_x$ thin film is the same as the bulk $TiO_2$, 4.32 g/cm$^3$. Both O and Ti are distributed about 10 nm from the surface, indicating that the elements in the thin film could be measured. Depth resolution at the interface between the thin film and the substrate is worse than that of the surface, which is considered to be the impact of multiple scattering when incident and recoil ions pass through the sample. An atomic concentration of O and Ti in the film is determined to be 77% and 23% from the average value between 2 and 6 nm. This corresponds to x = 3.40, and the concentration of Ti is lower than the value obtained by the RBS. The reason for the lower Ti concentration in the present TOF-ERDA measurement is probably due to the multiple scattering in the sample and carbon foil in T1. The deviation of the cross section from the Rutherford scattering due to the screening effect is expected to be small [6]. H and C can be seen on the surface, which may be due to contamination of the sample surface or a carbon build-up due to the beam irradiation. There may be a large uncertainty in the hydrogen concentration due to the low detection efficiency of hydrogen.

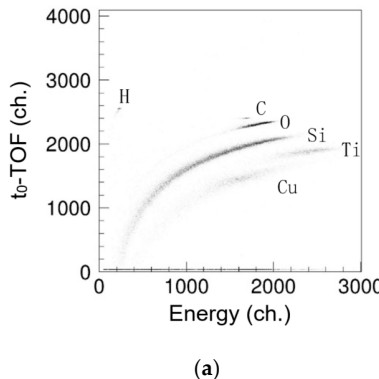 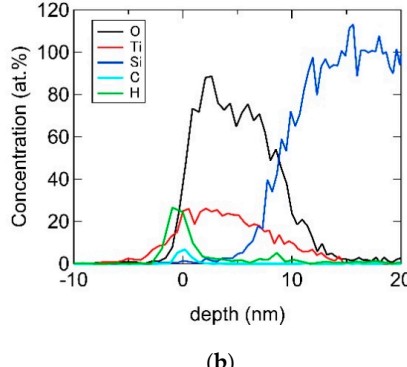

(**a**)          (**b**)

**Figure 14.** TOF-E two-dimensional plot (**a**) elemental depth profiling (**b**) $TiO_x$ thin film of about 10 nm thickness. The depth scale in nm is obtained by considering a film density of 4.23 $g/cm^3$.

## 6. Conclusions

In this review paper, the principle of the TOF-ERDA measurement and the measurement set-up, developed by the author, were described. The TOF-ERDA has been used for light element analysis and analysis of thin film samples at accelerator facilities around the world. In future works, we are considering combining TOF-ERDA with other methods such as TEM and XPS to analyze thin films.

**Funding:** This research was partially funded by the JSPS Grant-in-Aid for Scientific Research, grant number 19760621 and was supported by "Collaborative Research Project of the Wakasa Wan Energy Research Center."

**Acknowledgments:** The author gratefully appreciates Y. Kajitori, M. Oishi, H. Nakamura, Y. Haruyama, M. Saito, K. Suzuki, R. Ishigami, S. Hibi and M. Saito for their priceless help and fruitful discussions. The author is also grateful to the staff of the accelerator group at the Wakasa Wan Energy Research Center for the operation of the accelerator and conducting the TOF-ERDA measurements.

**Conflicts of Interest:** The authors declare no conflict of interest.

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
