# Peer review of "Time-Of-Flight ERDA for Depth Profiling of Light Elements"

_qubs, doi:10.3390/qubs4040040_

Round 1

Reviewer 1 Report

This paper is kind of a review manuscript related to the TOF-ERDA technique. It describes the basic physics behind the technique quite well (this info is quite general level) and it describes the measurement system the author has been using in his research and the performance of this setup. And this setup has the same performance as the instruments had already in 1980'ies.

What it lacks is the critical view required from a review article. The setup described here is one of the lower performance TOF-ERDA setups in the world when it comes to detection efficiency, TOF-resolution, sensitivity and depth resolution. There is easily 5-10 better instruments among about 15 in total in the world. The reader would get a false idea from this review paper that the described instrument is state-of-the-art which it is not.

This would be the major correction in this paper: to make it a real review paper in which also own performance is reflected to the performance of the others. For instance, there has been plenty of fairly recent developments concerning gas ionization detectors (GIC), which have replaced Si detectors in advanced setups but this has not been touched at all.

Another thing which disturbs me is the use of new results in a review paper. Here it seems that the sensitivity results (oxygen in Si) are the first time published here. The reported sensitivity is very modest one and one should get one or two orders of magnitude better sensitivities with TOF-ERDA for this sample type.  I would understand the approach if it would a review of the past work done especially with this instrument but this review is a more general one (although the references are mainly pointing towards the work by the author).

More minor issues are listed below:

1) line 41: It is false to say that TOF-ERDA is good for analyzing (only) light elements. It should be more stressed that heavy elements can also be measured and in most cases there is no need for RBS.

2) equation (3): It should be noted that this "average stopping approach" is a very coarse one and in better approach integrated energy losses should be used.

3) line 75: The non-Rutherford cross-sections are hardly relevant in TOF-ERDA in which heavier incident ions are used. It can be relevant in case of He beam. Screening is much greater problem in case of low-energy TOF-ERDA.

4) line 81: One cannot say that the number of scattered ions towards the detector is always higher than number of recoils. By choosing proper beam for proper sample the number of scattered ions can be (almost) zero). In order to reach better sensitivity this should be done.

5) line 94: There should be reference given here, I have seen much lower values as well.

6) line 110: In general, argon and krypton are hardly ever used but with Tandem accelerators Cl, Cu, Br, I and Au are much much more common beams.

7) line 128 and following pages: It is very misleading to use names of forward and backward detector for the TOF-detectors. Backward? Cannot understand the origin. A stand way is to use first and second time detector. This should be corrected.

8) line 134: The geometry used here is quite special when it comes to beam spot size (1 cm2) and defining aperture size (about 4 mm2). This in practice means that most of the recoils leaving the beam spot and going through the first detector never hit the second detector. The "active area" of the beam spot is much smaller than 1 cm2.

9) line 164: comment to the authors, the system Au/Si is hardly an optimal one for measuring the TOF resolution as Au does not form a very smooth layer on Si. A high quality alpha source would be a very good option.

10) line 174: The relevance of one detector time resolution is hardly of any relevance as you would always need two detectors to measure the time.

11) Figure 7: The reported detection efficiencies are low compared to other systems reporting higher than 99 % efficiency for carbon. The authors should somehow comment this. Can you even use the hydrogen in the analysis if the detection efficiency is below 10 % for relevant energies? Does it ever get to 100 %, with heavier ions? What kind of count rates were used when these were measured as detection efficiency is very sensitive to the count rate (or false counts in TOF1).

12) line 200 and Figure 8: The authors state here that efficiency curves measured for H, He and C are used to define the detection efficiency for other elements. This can be done but is hardly an elegant way. I would love to see the curve for some heavier element, like Si. This curve in Fig. 8 would predict that the detection efficiency never goes to 100 %. This can be the case in case the are holes in the foils but author does not say this. 

13) line 218: The depth resolution stated here would require one example fit to the front of the curve as an additional plot. To judge the quality of the fit. It should also be stated how the given resolutions are calculated. When giving the depth resolution in nm instead of at./cm2 one has to give the assumed density for the carbon film (and how it was determined).

14) line 224: Wording is wrong here. I agree that the main contribution is probably the kinematic effect (not mainly) but that is not self-evident from the plots as the effect of TOF-resolution also gets bigger with energy.

15) Sensitivity chapter: I think this is relevant and interesting part of the paper. In order to judge the quality of the measurement I request the author to give the count rates in T1 and E detectors and plot TOF-E histograms from the two measurements (pure Si and 16O implanted Si). And the plot should not be in pale grey scale which is very difficult to use to evaluate the data. The sensitivity is very sensitive to the count rate. The total number of coincident events as well as events in oxygen area (shown in these plots) should also be given. It would be even better to have four plot, two for Cu and two for C.

16) line 249 and surroundings: The authors claim that the sensitivity with Cu is higher (only) due to the higher cross-section when compared to C beam. This is hardly true. My bet is that sensitivity is better for Cu because there is much less background (probably huge C background) with Cu in oxygen area. If numbers are given for the sensitivity, there should also be uncertainties. Here I also point to the comment 8 above, the numbers of incident ions in the Fig. 10 are hardly relevant as only a small portion of the beams spot can even generate recoils which penetrate the T2 and reach E-detector.

17) Fig. 11 and corresponding text: What software was used to generate these depth profiles? This plot nicely show the limitation of the low-energy TOF-ERDA, the Ti depth profiles shows much lower concentration than 1:2 with respect to O. ANd this is most probably not true. The author should comment the reasons for this (screening, multiple scattering in the sample and carbon foil etc.). In b) the y-axis title is probably not density but concentration.  

Reviewer 2 Report

General comments

This manuscript presents the ToF-ERDA technique, which is a well-known ion beam analysis method that has been in use for over 30 years. The present description of the technique is in that sense not novel, nor very detailed compared to previous books and articles. The main merit with the article lies in the attempt to quantify the depth resolution and the sensitivity of the technique, and that it gives a clear (although brief) review of the technique with several key references. The paper is also well organized and comprehensively written and could be published. There are, however, a number of issues that needs to be considered before acceptance and these are listed below.

Specific comments

Row 35: Please, define “conventional ERDA”. Do you mean ERD(A) as opposed to ToF-ERDA?

Row 43: Presumably you discuss elastic Rutherford scattering.

Eq 2-4: Please, define x more clearly. I would guess that x is the straight distance travelled by the 1) incident ion and 2) the recoil in the material, since you multiply it with the stopping to get energy loss. However, in this case x will be different for case 1 and 2, if alfa and beta angles are different.

Row 65: The solid angle should be defined here. (Later it is explained that the solid angle is given by the first foil aperture.)

Row 112: It seems that cross section data is needed only for quantitative analysis of low mass ERDA and not heavy ion ERDA?

Row 125: Is there any benefits with a 5-axis goniometer, instead of for instance one with 3 axes?

Row 128, 129: Maybe you should use the more conventional terms “start” and “stop detector”, instead of “forward” and “backward time”.

Row 202 (for instance): If SRIM is used for Fig. 7, please state the version. If not, Please, give the source for stopping data.

Fig. 8: What happens for heavier ions than carbon? Is it some fundamental reason why you cannot reach the efficiency of 1?

Row 215: Please, describe the carbon better. Is it, for instance, diamond like or graphite?

Row 222: Explain “kinematical broadening”.

Figure 9 and text related to this: How is your depth resolution defined? This is a crucial point for the whole paper. From what I understand, you have 50 nm of pure carbon, so how can you know what depth a particular carbon ion is coming from. Maybe you could also explain why the dependence on the incident and exit angles? Is there an optimal combination for best depth resolution?

Eq. 11: Please, give a reference, or a thorough motivation for this expression.

Fig. 10: Why should the sensitivity, defined in Eq. 11, drop for larger number of incident ions?

Fig. 11: The y-scale (density (au)) is strange. It suggests that the density of the Si bulk is the same as the O density in the film, and the density of Ti in the film is about one third of the O. The hydrogen peak is also comparatively large. Any reason for this? Furthermore, it should be possible to convert the arbitrary units to atomic percent. Please, comment.

Reference 22: This reference has a clear overlap with the present article. Is it true that the data presented in this new article has not been published before? (I could just see the abstract from this reference, but it seemed to be very similar to some of the things mentioned in the new article.)

Round 2

Reviewer 2 Report

The paper has been extensively reviewed and expanded and I think it is ready for publication. I only have one minor question about Eq. 13. It would be helpful if the authors could include a reference for this equation, or present some other motivation.
